# On the difficulty of producing good linguistic lies

Lucía Catalán Gris[1]    Kim Gerdes[1]

(1) Université Paris-Saclay, Lisn, CNRS, 91400 Orsay, France
`lucia.catalan-gris@lisn.fr, kim.gerdes@lisn.fr`

## RÉSUMÉ

Tester si les grands modèles de langage (LLM) capturent la structure linguistique (au-delà de la fluidité de surface) nécessite des affirmations linguistiques vraies et fausses. Bien que les affirmations vraies puissent être tirées de la littérature, produire des équivalents faux convaincants (« mensonges linguistiques ») n'est pas trivial. À l'aide de 235 paires d'affirmations tirées d'articles de syntaxe théorique, nous testons GPT-5.2 et observons une forte asymétrie : il rejette efficacement les affirmations fausses (spécificité élevée) mais confirme difficilement les affirmations vraies (faible rappel). Nous examinons trois causes possibles — les indices de surface, le type de contradiction (négation simple vs contradiction sémantique) et le format de présentation (isolé, par paires, en lots) — et montrons que (i) les indices de surface n'expliquent pas cet effet, (ii) la négation explicite n'apporte aucune amélioration, et (iii) le formatage contrastif/par paires gonfle les performances. Enfin, le modèle présente un biais de rejet persistant (48% de faux négatifs sur les affirmations vraies), ce qui suggère que cette asymétrie est davantage dictée par la prudence que par un manque de connaissances.

## ABSTRACT

**On the difficulty of producing good linguistic lies**

Testing whether Large Language Models (LLMs) capture linguistic structure (beyond surface fluency) requires true and false linguistic claims. While true statements can be taken from the literature, producing convincing false counterparts ("linguistic lies") is non-trivial. Using 235 statement pairs from theoretical syntax articles, we test GPT-5.2 and find a strong asymmetry : it rejects false claims well (high specificity) but confirms true ones poorly (low recall). We probe three possible causes—surface cues, contradiction type (simple negation vs. semantic contradiction), and presentation (isolated, pairwise, batch)—and show that (i) surface cues do not explain the effect, (ii) overt negation does not help, and (iii) contrastive/pairwise framing inflates performance. Finally, the model shows a persistent rejection bias (~48% false negatives on true statements), suggesting that caution, more than lack of knowledge, drives the asymmetry.

MOTS-CLÉS : négation, génération de contradictions, vérification des connaissances linguistiques, évaluation de grands modèles de langue, analyse d'articles scientifiques, évaluation zero-shot.

KEYWORDS: Negation, Contradiction Generation, Linguistic Knowledge Verification, Large Language Model Evaluation, Analysis of Scientific Articles, Zero-shot Evaluation.

# 1   Introduction

Large Language Models (LLMs) demonstrate impressive language fluency ; however, evaluating their deep understanding of language remains a significant challenge (Mahowald *et al.*, 2024). Because

LLMs operate fundamentally as statistical prediction engines, standard performance benchmarks often fail to reveal whether they genuinely comprehend the underlying structures of human linguistic cognition (Katzir, 2023). To rigorously determine whether these models possess structured linguistic knowledge rather than relying on surface-level heuristics, it is necessary to assess their ability to distinguish true from false linguistic facts. However, constructing high-quality linguistic "lies", i.e. statements that logically contradict established grammatical or syntactic facts without being trivially obvious, presents a fundamental methodological challenge.

We use the dataset from (Catalán-Gris *et al.*, 2026) : 235 manually curated statements from 11 theoretical syntax articles from Glossa. In that study, GPT-5.2 performed best overall (75% accuracy), and its self-reported confidence correlated with correctness. Although the overall accuracy is relatively high, the precision (68%) compared with recall (52%) and specificity (87%) suggests that GPT-5.2 is better at rejecting false statements than at confirming true ones.

To better understand this phenomenon and isolate its source, we investigate three potential factors. The research questions are :

**1. Are performance differences driven by shallow cues ?**

One possibility is that true and false statements differ systematically in surface form, which may allow models to rely on shallow cues rather than genuine linguistic knowledge. Simple baseline classifiers, such as logistic regression, are trained to determine whether language models exploit stylistic cues.

**2. Does contradiction complexity affect LLM accuracy in Linguistic Knowledge Evaluation ?**

Although most false statements do not contain explicit negation, certain contradictions may involve implicit or structural forms of it. Therefore, model performance is evaluated on contradictions both with and without overt negative markers to determine whether it contributes to the observed asymmetry.

**3. Does the structural presentation of linguistic claims influence the LLM's linguistic knowledge ?**

This study investigated whether the structural presentation of linguistic claims affects GPT-5.2's zero-shot Linguistic Knowledge Evaluation capabilities. Specifically, the performance when prompted to evaluate statements in isolation, in lists (batch processing), or as pairwise comparisons by selecting the more "accurate" statement.

Our main contributions are : (i) a baseline analysis showing that surface cues do not explain LLM performance on this task, (ii) a controlled comparison of simple syntactic negation vs. complex contradiction that reveals divergent model behaviour, and (iii) an evaluation of structural presentation effects (list-based vs. pairwise) on zero-shot linguistic fact-checking.

# 2 Task Definition and Motivation

The objective of **Linguistic Knowledge Evaluation** is to determine whether LLMs possess authentic linguistic knowledge rather than relying solely on surface-level pattern recognition. Assessing performance on this task requires both true and false linguistic statements. If statements published in peer-reviewed journal articles are regarded as "true", then it is necessary to construct corresponding false statements that contradict them for comprehensive evaluation.

## 2.1 Related Work

Previous research has extensively documented the difficulties LLMs face when processing negation. Hosseini *et al.* (2021) and Truong *et al.* (2023) demonstrate that despite their general fluency, language models systematically struggle to comprehend and accurately handle negative contexts. Hosseini *et al.* (2021) conclude that standard pre-trained language models often mishandle negation, and that adding a negation-based unlikelihood objective can improve them. Contrarily, Truong *et al.* (2023) evaluated GPT-neo, GPT-3, and InstructGPT across a wide range of negation benchmarks and found "several limitations including insensitivity to the presence of negation, an inability to capture the lexical semantics of negation, and a failure to reason under negation". This vulnerability extends to complex metalinguistic tasks, where LLMs fail to exhibit robust linguistic knowledge (Begus *et al.*, 2023), and it significantly exacerbates hallucination rates in tasks involving negated statements (Varshney *et al.*, 2025). Consequently, standard evaluation frameworks often fall short, prompting researchers to develop negation-aware metrics for a more rigorous assessment of language generation systems (Anschütz *et al.*, 2023). Beyond evaluation, testing these models requires high-quality contrastive examples. To generate such contradictory claims, prior approaches have used targeted semantic edits on existing sentences (Hidey & McKeown, 2019) or combined LLMs with explicit linguistic rules to generate robust prototypes for contradiction detection (Pielka *et al.*, 2023).

## 2.2 On Defining Contradiction and its Problems

At its most basic level, a contradiction is defined as the logical incompatibility between two or more propositions. In the context of propositional calculus, it is most frequently expressed as the conjunction of a statement ($P$) and its denial ($\neg P$), resulting in the formula : $P \land \neg P$ (Horn, 2024). Varzi (2004) notes that while this formula represents a single contradictory statement, "a contradiction can also be understood as a pair of sentences where one is the direct negation of the other". Within a truth table, a contradiction is identified as a statement that is false under all possible interpretations of its constituent components.

However, this propositional framework is a starting point rather than a complete model for our task. Natural language linguistic claims frequently involve quantifiers ("usually", "some", "most"), modalities ("may favor", "can achieve"), and hedging, which resist reduction to simple $P/\neg P$ pairs. Moreover, in classical logic, a distinction is drawn between *contradictories* (exactly one must be true : $P$ vs. $\neg P$) and *contraries* (both can be false : "all X are Y" vs. "no X are Y"). Many of our complex contradictions are, strictly speaking, contraries rather than contradictories : they assert an incompatible alternative rather than a strict logical negation. We adopt this broader notion deliberately, since strict $\neg P$ negations of gradient linguistic generalizations are often trivial or pragmatically unnatural.

In our case, let S be the statement that "In Basque, agentive verbs usually occur with an ergative subject and the auxiliary edun" extracted from (Pineda & Berro, 2020). Its contradiction should be formulated as "In Basque, no agentive verbs occur with an ergative subject and the auxiliary edun".

However, a contradiction in natural language is often more ambiguous. Rather than simply indicating that a statement is false, negation can act as a "metalinguistic" tool to reject a claim because of its wording, pronunciation, or possible misleading effects (Horn, 2024). As a result, the negation of a linguistic generalization is not always straightforward to interpret.

Evaluating linguistic knowledge requires generating contradictions for linguistic statements, which

poses a unique set of challenges. Unlike domains with rigid factual boundaries, linguistics is characterized by cross-linguistic variation, dialectal nuance, and competing theoretical frameworks. If we use the earlier example and try to negate S by saying "In Basque, agentive verbs usually occur with an ergative subject and the auxiliary izan," it is not clear if this really creates a $\neg S$ statement, because in Basque grammar, *izan* (the intransitive auxiliary) is not simply the opposite of *edun* (the transitive auxiliary)—they are complementary within the split-intransitivity system rather than logical antonyms. The incompatibility is not exact, and the resulting statement is a contrary rather than a contradiction.

## 2.3 Contradiction Generation Problems

To illustrate these challenges, several cases are presented in which GPT-5.2 mishandled false statements from the dataset :

1. **Over-acceptance of Partial Truths :** GPT-5.2 shows considerable leniency toward partial truths. For the false claim "In Samoan, some intransitive verbs combine with an ergative subject," the model rationalized a "True" verdict by citing "split/optional ergative marking," despite the gold standard treating Samoan intransitive subjects as strictly absolutive. The contradiction is well-formed ; the failure lies in the model's overly lenient reasoning.
2. **Vulnerability to Hedged Terminology :** Hedging is pervasive in linguistics (Livytska, 2019), and retaining hedged language in contradictions often prevents a "False" judgment. For the claim "The strategies to express verum in Spanish, English, and German appear related on a superficial level," the model noted "the statement is non-specific, so it cannot be falsified easily." This is partly a contradiction-quality problem : hedged claims resist falsification by design.
3. **Theoretical Apologism and Hallucinations :** Rather than citing literature, the model derives answers from first principles, supporting false claims (Augenstein *et al.*, 2023). For the false statement "Minimize Domains (MiD) may favor P-omission," it hallucinated : "omitting a preposition can reduce domain size [...], so MiD can favor P-omission." The contradiction is well-constructed, but the model generated a fictitious justification.
4. **Overgeneralization of Psycholinguistic Findings :** GPT-5.2 relies on heuristic knowledge rather than evaluating specific experimental parameters. For contradictory claims about L2 anaphora resolution processing costs, the model accepted false statements by citing the generalization that "many studies report measurable online processing costs," ignoring that the specific experimental condition was empirically false.

These examples show that negation is more than just a marker to reverse. It interacts with scope, presupposition, and changes in meaning in ways that LLMs often find difficult to handle.

# 3 Experimental Design

## 3.1 Dataset

In the paper (Catalán-Gris *et al.*, 2026) [1], we generated two false variants for each statement : one using GPT-5.2 and one using Gemini 2.5 Flash. To this end, we used a contradiction prompt that

---

1. All data, prompts, and code are publicly available under CC-BY 4.0 at https ://github.com/linguistic-fact-checking/nslp26

required the model to generate a strict logical contradiction without using negative markers such as "no" or "not", on the assumption that overt negation would be too easy for an LLM to detect as a surface cue.

For these experiments, we manually reviewed the negation variants and constructed a gold-standard negation to balance the data, resulting in 470 statements : 235 true and 235 false. From now on, we will refer to these false statements as "complex contradictions"(1c).

To create "simple negations" (1b), we employed Negate [2], a Python module library for sentence negation that focuses on verbal negations (Anschütz *et al.*, 2023). It uses spaCy's *en_core_web_md* model for part-of-speech tagging and dependency parsing. This automatic system generated simplified negations by applying standard syntactic transformations (e.g., inserting "don't" in appropriate positions). Given the technical nature of linguistic terminology, 7 negations required manual correction to preserve semantic accuracy and naturalness.

(1)    a. In Warlpiri, all intransitive verbs combine with an absolutive subject. [true] (extracted from Pineda & Berro (2020))

b. In Warlpiri, intransitive verbs do not combine with an absolutive subject. [false ; simple negation]

c. In Warlpiri, all intransitive verbs combine with an ergative subject. [false ; complex contradiction]

## 3.2   Are performance differences driven by shallow cues ?

We trained supervised classifiers on the same statements. If simple models that rely primarily on surface-level features (such as word patterns) perform well, this would suggest that the dataset has shallow cues that can be exploited. In such cases, the language model's performance may be sensitive to these cues rather than reflecting genuine reasoning.

**Dataset.** The first step was to create a balanced set of linguistic statements, categorized as positive or negative based on the presence or absence of negative markers. Negatives are not common in the data. Out of 470 statements, only 50 have them, which is 10.6%. We collected 88 samples, with 22 for each combination, for the final dataset.

**Feature Representations.** We tested three feature representations, word-level TF-IDF vectors, sentence-level embeddings from SPECTER (Cohan *et al.*, 2020), and contextual embeddings from DistilBERT (Sanh *et al.*, 2019).

**Models.** We evaluated three classical machine learning classifiers : Logistic Regression, Naive Bayes, and Support Vector Machine (SVM), as well as a transformer-based model, DistilBERT.

**Evaluation Metrics.** We employed $k$-fold cross-validation with $k \in \{3, 5\}$ to evaluate model performance based on accuracy, precision, recall, specificity, and F1-score.

---

2.  https ://github.com/dmlls/negate

## 3.3 Does contradiction complexity affect LLM accuracy in Linguistic Knowledge Evaluation?

The primary aim of this experiment was to isolate and evaluate the impact of simple structural negation on the veracity judgments of GPT-5.2. Specifically, we examine whether the simple negations, as opposed to the complex contradictions, affect model accuracy and confidence calibration.

**Datasets.** There are two versions : one named "With Simple Negation," which contains 235 true statements along with their 235 simple negations ; and another called "With Complex Contradiction," which includes 235 true statements and 235 complex contradictions.

**Prompt design.** We used a strict, zero-shot prompting strategy setting the LLM in the role of a "Linguistic Fact-Checker," requiring it to output a boolean verdict, a confidence score (0.0-1.0), and a rationale (see Appendix A).

**Evaluation.** We aggregated broad performance metrics : accuracy, precision, recall, specificity, and F1-score. To evaluate confidence calibration, we implemented a threshold-based filtering approach and calculated the corresponding Accuracy and Coverage (the proportion of total statements meeting the threshold criteria).

## 3.4 Does the structural presentation of linguistic claims influence the LLM linguistic knowledge?

To investigate whether the framing and structural presentation of linguistic claims influence GPT-5.2's zero-shot Linguistic Knowledge Evaluation capabilities, we designed an evaluation framework comprising two distinct prompting paradigms (see Appendix A for full prompt templates).

**Paradigm A : Pairwise Contrastive Evaluation**

In the first experimental paradigm, we framed the task as a comparative analysis. GPT-5.2 was presented with two concurrent statements (Statement A and Statement B) and instructed to determine which claim was more theoretically sound and empirically accurate based on modern descriptive linguistics. The model was required to output a structured JSON object that specifies its preferred statement, individual truth verdicts for A and B, a brief linguistic rationale, and a confidence score.

To evaluate the model's discriminative robustness and isolate potential confounders, such as positional bias and stylistic complexity, we tested the following configurations :

— True vs. False (T/F & F/T) : We paired true linguistic statements, such as (1a), with their corresponding false counterparts (1b, 1c). To control for positional bias, we systematically alternated the order in which the empirically valid claim appeared, as Statement A or Statement B. We evaluated the original true statements against the complex contradictions and the simple negations.

— False vs. False (F/F) : We paired two erroneous statements against each other. This included contrasting a complex contradiction with a simple negation of the same root claim to observe text-style preferences, as well as pairing completely unrelated false statements. For these pairs, "accuracy" is defined as the proportion of trials in which the model preferred the complex contradiction over the simple negation, since the complex contradiction, being LLM-generated, is a more plausible falsehood, and thus the harder test case.

(2)  a. Spanish verum cannot stem from sentence mood. [false ; simple negation] (original extracted from Kocher (2023))
b. Embedded gapping is ungrammatical in Spanish. [false ; complex contradiction] (original extracted from Bîlbîie & de la Fuente (2019))

— True vs. True (T/T) : As a baseline negative control, we prompted the model to compare two unrelated, factually accurate statements. Since both statements are true, there is no correct answer ; we report the preference distribution (proportion choosing A vs. B) as a measure of positional bias, with a 50/50 split representing the ideal baseline.

(3)  a. The clausal coordinator sì in Yorùbá has an unusual surface position in the middle field of the second conjunct. [true] (original extracted from Aremu & Weisser (2024))
b. Negation is a universal property of language. [true] (original extracted from Childs (2017))

**Paradigm B : Batch List Evaluation**

In the second paradigm, we shifted from a contrastive framework to a batch-evaluation format. The model was presented with lists of 10 independent linguistic statements from the dataset that contains true statements and complex contradictions. The prompt explicitly instructed the model to evaluate each statement independently, prohibiting cross-contamination or relative baseline comparisons between claims in the list.

# 4    Results and Discussion

## 4.1    Are performance differences driven by shallow cues ?

**Results :** Overall, the classification results remain close to a random baseline across all configurations, indicating that stylistic features are poor predictors of the verification labels. The highest performance was achieved with a simple TF-IDF representation paired with a Logistic Regression classifier, yielding an accuracy of 51.2% and an F1-score of 53.7% (with $k = 3$). Notably, more complex, context-aware approaches, such as the fine-tuned DistilBERT model, struggled to generalize and underperformed simpler traditional baselines, achieving only 44.4% accuracy.

**Discussion :** The consistent inability of the models to accurately classify statements based on their lexical or embedding representations suggests that sentence style—including surface-level negation markers—does not inherently encode the veracity of the statements and is thus unlikely to be the primary factor driving LLM responses.

## 4.2    Does contradiction complexity affect LLM accuracy in Linguistic Knowledge Evaluation ?

**Results :** As can be seen in Table 2, the use of simple negation did not simplify the task. GPT-5.2 achieved a slightly higher accuracy (72.1%) and F1 score (65.1%) when evaluating complex contradictions than it did on the simplified negations (70.2% accuracy, 63.4% F1 score). These

| Feature | Model | $k$ | Accuracy | F1-Score | Precision | Recall | Specificity |
|---------|-------|-----|----------|----------|-----------|--------|-------------|
| DistilBERT | Transformer | 3 | 0.444 | 0.316 | 0.291 | 0.356 | **0.588** |
| | | 5 | 0.375 | 0.344 | 0.369 | 0.381 | 0.412 |
| Specter | LG | 3 | 0.420 | 0.474 | 0.436 | 0.524 | 0.319 |
| | | 5 | 0.430 | 0.400 | 0.434 | 0.383 | 0.475 |
| | NB | 3 | 0.443 | 0.465 | 0.443 | 0.497 | 0.384 |
| | | 5 | 0.410 | 0.416 | 0.410 | 0.433 | 0.389 |
| | SVM | 3 | 0.500 | 0.469 | **0.652** | 0.549 | 0.467 |
| | | 5 | 0.432 | 0.527 | 0.452 | **0.639** | 0.231 |
| TF-IDF | LG | 3 | **0.512** | **0.537** | 0.519 | 0.567 | 0.459 |
| | | 5 | 0.431 | 0.398 | 0.411 | 0.428 | 0.439 |
| | NB | 3 | 0.431 | 0.406 | 0.458 | 0.384 | 0.484 |
| | | 5 | 0.451 | 0.390 | 0.434 | 0.361 | 0.547 |
| | SVM | 3 | 0.455 | 0.496 | 0.484 | 0.543 | 0.371 |
| | | 5 | 0.429 | 0.417 | 0.420 | 0.450 | 0.411 |

TABLE 1 – Classification performance across different combinations of feature representations (TF-IDF, Specter embeddings, and a fine-tuned DistilBERT) and machine learning models

results indicate that the presence of explicit syntactic negation markers does not improve the model's Linguistic Knowledge Evaluation capabilities ; rather, it is barely more adept at properly classifying the LLM-generated contradictions in the original dataset.

| | Accuracy | F1-Score | Precision | Recall | Specificity |
|---|----------|----------|-----------|--------|-------------|
| With Simple Negation | 0.702 | 0.634 | 0.823 | 0.515 | 0.890 |
| With Complex Contradiction | **0.721** | **0.651** | **0.871** | **0.520** | **0.923** |

TABLE 2 – GPT-5.2 performance on Linguistic Knowledge Evaluation by type of false statements in the dataset.

Confidence calibration shows consistent improvement as thresholds increase for both negation types : at the 0.75 threshold, accuracy reaches 82.68% for complex contradictions and 84.21% for simple negations—a difference of 1.53 percentage points that is not statistically significant ($\chi^2 = 0.0580$, $p = 0.8097$). This suggests that GPT-5.2's ability to map confidence to correct judgments is similarly effective across negation types.

**Discussion :** The performance discrepancy between the two dataset variations suggests that the model does not rely on superficial negative markers. Instead, GPT-5.2 is more responsive to broad semantic incompatibility. Natural, complex contradictions often employ antonyms or structurally divergent propositions, which arguably create a larger distance in the model's vector space than the localized insertion of a single negative particle.

However, the model exhibited a strong bias toward predicting statements as false, resulting in a 48% false-negative rate on true statements. This indicates that, while semantically rich contradictions are marginally easier for the model to interpret than simple negations, the claim's structural phrasing is largely eclipsed by the model's baseline hesitance to validate positive linguistic assertions.

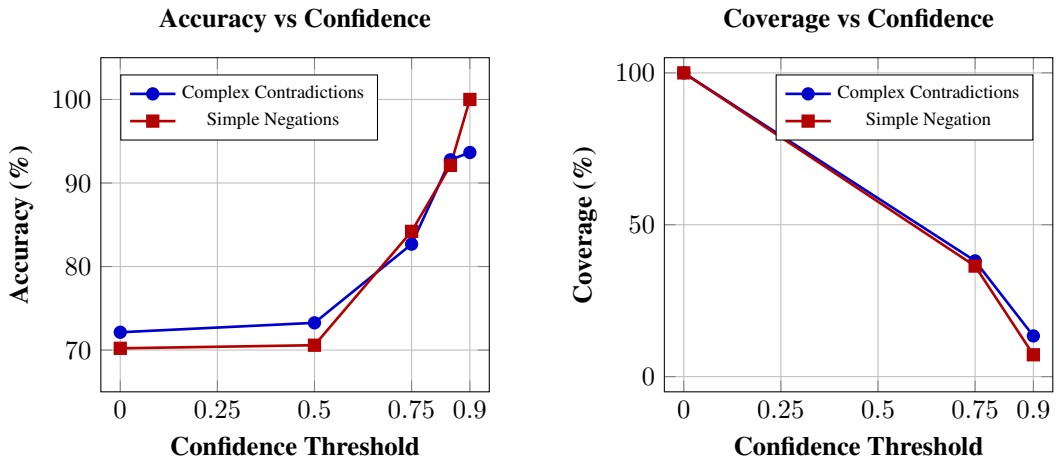

FIGURE 1 – Comparative analysis of model accuracy (left) and coverage (right).

Confidence filtering improves accuracy for both negation types, but the scores are self-reported and should be read as ordinal uncertainty signals rather than calibrated probabilities.

## 4.3 Does the structural presentation of linguistic claims influence the LLM linguistic knowledge ?

**Results :** When evaluating statements presented as long lists, our model achieved an overall accuracy of 74.68%, displaying a solid ability to fact-check linguistic claims independently. The specificity score (80.9%) outpaced the precision (78.16%) and recall (68.51%) scores, indicating the model was marginally less likely to endorse a true statement as factual, but highly reliable when it did. The final F1-score stood at 73.02%. In general, the performance of GPT-5.2 was higher compared to sending isolated claims (72%) as in Section 4.2.

However, when shifted to a Pairwise Comparison setting, presenting a true statement directly against either a complex contradiction or a simple negation shifted the model's reliability based primarily on the type of comparison and the positioning of the correct answer.

When the true statement was presented first (Statement A), the model consistently preferred it over the false alternative, demonstrating strong discrimination abilities. Conversely, when the false statement was positioned first (Statement A = False, Statement B = True), the performance degraded slightly. This distinct drop indicates a minor "primacy bias", in which the model slightly anchors on the first statement it encounters, making it harder to reject an initially plausible-sounding false claim purely on mathematical grounds.

To measure robustness against structural deception, we forced the model to compare two false statements (F/F) or two unrelated true statements (T/T). When presented with two false statements (e.g., Simple Negation vs Complex Contradiction), the model showed clear uncertainty. Instead of solidly rejecting both, it leaned heavily on stylistic preferences—favoring the simple negation 56% to 62% of the time, suggesting that syntactic simplicity might masquerade as correctness to the LLM. When evaluating two true statements (T/T), it struggled to balance the scales, showing an unbalanced

preference split (favoring A 39%, B 61%) instead of an equal equilibrium.

| A | B | Accuracy | Recall | F1-score |
|---|---|---|---|---|
| True statement | Complex Contradiction | **0.809** | **0.809** | **0.894** |
| Complex Contradiction | True statement | 0.783 | 0.783 | 0.878 |
| True statement | Simple Negation | 0.770 | 0.770 | 0.870 |
| Simple Negation | True statement | 0.732 | 0.732 | 0.845 |
| Complex Contradiction | Simple Negation | 0.621 | 0.621 | 0.766 |
| Simple Negation | Complex Contradiction | *0.421* | *0.421* | *0.593* |
| Complex Contradiction | Complex Contradiction | 0.604 | 0.604 | 0.753 |
| True statement | True statement | 0.608 | 0.609 | 0.757 |

TABLE 3 – GPT-5.2 performance on Linguistic Knowledge Evaluation by pairwise Contrastive Evaluation.

**Discussion :** It is important to note that the pairwise paradigm reformulates verification as a *discrimination* task rather than a classification task : the model must only identify the better of two options, which is inherently easier than evaluating a claim in isolation. The higher absolute accuracies in the pairwise setting should therefore not be directly compared to the single-statement or list results. With this caveat in mind, these findings explicitly demonstrate that while GPT-5.2 can fact-check complex linguistic data, its structural presentation heavily modifies performance. The Independent List Paradigm mitigates comparison bias but suffers from lower recall, indicating the model is more hesitant to confirm truths without relative context. The Pairwise Paradigm yields stronger recognition of facts when the true premise is presented before an opposing argument. However, forcing comparative reasoning induces a forced-choice fallacy when both claims are equally true, leading the model to hallucinate "superiority" based on style or grammar rather than linguistic fact.

# 5   Conclusion

We investigated the difficulty of verifying linguistic knowledge in LLMs by evaluating GPT-5.2's ability to discern established theoretical facts from synthetic linguistic "lies". Our three experiments show that (i) surface-level classifiers hover near chance (44–51%), confirming that the task requires deep semantic reasoning ; (ii) explicit negation markers do not improve model accuracy—complex contradictions involving antonyms or structural divergence are marginally easier to detect than simple syntactic negations ; and (iii) structural presentation strongly modulates performance, with pairwise contrastive framing inflating accuracy relative to list-based or isolated evaluation. Overall, while GPT-5.2 possesses some theoretical linguistic knowledge, robust evaluation requires carefully calibrated, complex "lies."

**Limitations.** We evaluate one model (GPT-5.2) on one journal source (Glossa) with a small dataset (235 pairs ; 88 samples for RQ1), in English and zero-shot. We did not significance-test the small RQ2 difference (70.2% vs. 72.1%) or vary decoding/prompt settings ; moreover, linguistic "truth" is framework-dependent and negating quantified generalizations can be pragmatically marked.

Future work should address these limitations by testing open-source LLMs and multilingual datasets, exploring few-shot prompting and fine-tuning strategies, scaling the dataset to other linguistic subfields (phonology, semantics, pragmatics), and developing human-in-the-loop pipelines for generating more controlled, theoretically grounded false statements.

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

# A   Prompts used for the tasks

## A.1   Linguistic Knowledge Evaluation Task

Role : Linguistic Fact-Checker (Syntax/Morphosyntax).
Task : Evaluate the accuracy of STATEMENT based on descriptive academic linguistics.
Return only JSON :
{
"verdict" : true or false,
"confidence" : 0.0-1.0,
"rationale" : "concise reason"
}
RULES :
- No prescriptive grammar ; use descriptive evidence.
- If the statement is a single word or lacks propositional content, the verdict is false.

## A.2  Pairwise Contrastive Evaluation Task

Role : Linguistic Fact-Checker (Syntax/Morphosyntax).
Task : Compare STATEMENT A and STATEMENT B against descriptive academic linguistics.
Determine which statement is more accurate or theoretically sound.
Return only JSON :
{
"preferred_statement" : "A" or "B",
"verdict_A" : boolean,
"verdict_B" : boolean,
"comparison_rationale" : "Identify why one is superior or why both fail.",
"confidence" : 0.0-1.0
}
Evaluation Rules :
Descriptive Baseline : Use empirical evidence (how language is actually structured/used) rather than
prescriptive "school" rules.
Strict Formatting : Return only the JSON object. No prose introduction or conclusion.

## A.3  Batch List Evaluation

Role : Linguistic Fact-Checker.
Task : You will be given a LIST of linguistic statements. Evaluate each statement independently as
if reviewing separate claims in an academic paper. Determine whether the statements are factually
and/or theoretically accurate in modern linguistics.
Evaluation Principles :
1. Use modern descriptive linguistics and typological consensus.
2. Allow framework-dependent interpretations (e.g., generative vs functionalist linguistics).
3. Do NOT evaluate style or grammar unless it affects meaning.
4. Each statement must be evaluated independently (no cross-contamination).
5. Focus on linguistic theory and empirical cross-linguistic evidence.
Descriptive Baseline : Use empirical linguistic evidence (attested language structures and typological
generalizations), not prescriptive grammar rules.
Strict Formatting : Return only the JSON object. No prose introduction or conclusion.
Input Format : You will receive : {
"statements" : ["statement 1", "statement 2", "statement 3", ...]
}
Return only JSON :
{
"results" : [ { "id" : 1,
"statement" : statement_text,
"verdict" : boolean,
"confidence" : 0.0-1.0,
"rationale" : "detailed explanation grounded in linguistic reasoning",
"evidence" : "linguistic generalizations or known typological facts"
} ] }