# OpenReview forum: "On the difficulty of producing good linguistic lies"
_ls2n.fr/CORIA-TALN/2026/Workshop/ARTS — ls2n CORIATALN 2026 Workshop ARTS Submission_

### Official Review · Reviewer_ggVA · 2026-04-30

**Mode De Presentation:** Poster

**Confience:**

Non

**Decision:**

Accepté

**Relecture:**

De manière générale, ce travail se situe en dehors de mon domaine d’expertise, et j’ai eu des difficultés à bien cerner son propos ainsi que ses contributions principales. Mon évaluation porte donc surtout sur sa pertinence pour l’atelier. De ce point de vue, le travail me semble entrer dans les thématiques d’ARTS, dans la mesure où les données utilisées sont extraites d’articles scientifiques.

**Resume:**

Cet article explore la capacité des LLMs à détecter des affirmations linguistiques vraies ou fausses. À partir d’un jeu de données constitué de paires d’énoncés, faits et contradictions, issus d’articles de syntaxe théorique, les résultats préliminaires montrent une asymétrie dans les performances des modèles, qui semblent avoir davantage de difficulté à identifier les affirmations vraies. L’article examine trois sources potentielles de ce problème. Les résultats suggèrent que les modèles exploitent certains indices de surface, que les négations explicites n’améliorent pas nécessairement les performances, et qu’une présentation contrastive des énoncés, sous forme de paires, tend à gonfler les résultats. Les auteurs observent également un biais des modèles vers les faux négatifs.

---

### Official Review · Reviewer_m67v · 2026-05-04

**Mode De Presentation:** Oral

**Confience:**

Oui

**Decision:**

Accepté

**Relecture:**

Strengths

- The contribution is interesting and well motivated, with a good focused question.
- There is a good effort to isolate possible theoretical reasons for the behaviour and the question is handled with reasonable care

Weaknesses

- The article could in my opinion have benefited from some more examples to help truly understand the types of examples and predictions of the model.
- There is no discussion about the fact that GPT was used to help generate the false statements and that the same model is used to evaluate - what effect could this have?

Comments

- There are some examples given inline in the text. However, it would have been helpful to have had some lists of examples to illustrate true and false statements. This would help the problem seem a little less abstract to me.
- There is no discussion over the fact that the model being analysed (GPT-5.2) is one of the models that was used to generate one of the false variants of statements. Does this have an effect? Is there any difference in the behaviour when applied to the two sets of false statements?
- I am not convinced by the framing of the initial experiment (whether performance is driven by shallow cues). Supervised classifiers are trained and if they pick up on surface-level features, this suggests the dataset has such a bias. However the link with LLM performance is still speculative in my opinion. Perhaps the LLM is picking up cues that are not captured by the simpler classifiers, or maybe it won’t behave in the same way faced with those surface features. I still think the experiment is valid, but I wouldn’t say the initial sentence “To determine whether GPT-5.2 predictions rely on shallow cues…”.

Typos, formatting, etc.

- Mots clés: “modèles linguistiques” -> “modèles de langue”
- There is an additional space before the ; and : (should not be there in English)
- I personally find the structure of Section 2 a  little odd (with the related work paragraph in the introduction to the section). A more traditional structure would see the related work extracted into its own section I think, and it could potentially be expanded a little beyond the narrow (but very relevant) current scope.
- Hosseini et al. (2021) concludes -> … conclude
- I recommend curly quotes rather than straight ones (i.e. ``…’’ in latex)
- Utilized -> maybe a personal thing, but I always recommend avoiding this in favour of the simpler “use”, as it is often used when use it perfectly fine (and it tends to be overused by AI)
- extracted from Pineda & Berro (2020) -> … from (Pineda & Berro, 2020)

**Resume:**

The topic of the article is investigating the knowledge LLMs have of linguistic structure, tested by looking at their performance on true and false statements about linguistic structure.

The article is based on a previous work by the same authors, where they investigated model performance on a curated dataset of 235 examples (taken from articles on theoretical syntax) with two generated false statements for each example. They investigate some of the findings of that paper, i.e. it detects false claims easily but has low recall on true claims and the fact that when it was highly confident it was often right.

They do this by exploring 3 questions:
1. Is the behaviour due to difference in the surface forms of true and false statements?
2. Does the complexity of the contradictions affect performance?
3. Does the way in which examples are presented have an influence (I.e. in isolation or several presented as a single list)?

They analyse GPT-5.2 predictions and behaviours. The findings suggest that (i) there are not that many easily detectable surface-based cues, (ii) the model performance does not seems to be affected by complexity of negation and (iii) performance does change considerably when examples are presented as lists, and there is a bias to preferring the first of two examples presented.

---

### Decision · Program_Chairs · 2026-05-07

Accept (Poster)